# Reply to Otter et al. Comment on “Bernard et al. Association between Urinary Metabolites and the Exposure of Intensive Care Newborns to Plasticizers of Medical Devices Used for Their Care Management. *Metabolites* 2021, *11*, 252”

**DOI:** 10.3390/metabo11090598

**Published:** 2021-09-03

**Authors:** Lise Bernard, Yassine Bouattour, Morgane Masse, Benoît Boeuf, Bertrand Decaudin, Stéphanie Genay, Céline Lambert, Emmanuel Moreau, Bruno Pereira, Jérémy Pinguet, Damien Richard, Valérie Sautou

**Affiliations:** 1Université Clermont Auvergne, Clermont Auvergne INP, CNRS, CHU Clermont Ferrand, ICCF, F-63000 Clermont-Ferrand, France; lise.bernard@uca.fr (L.B.); ybouattour@chu-clermontferrand.fr (Y.B.); 2Université de Lille, CHU Lille, ULR 7365 GRITA, F-59000 Lille, France; morgane.masse@univ-lille.fr (M.M.); bertrand.decaudin@univ-lille.fr (B.D.); stephanie.genay@univ-lille.fr (S.G.); 3CHU Clermont-Ferrand, Service Réanimation Pédiatrique et Périnatalogie, F-63000 Clermont-Ferrand, France; bboeuf@chu-clermontferrand.fr; 4CHU Clermont-Ferrand, Direction de la Recherche Clinique et de l’Innovation, F-63000 Clermont-Ferrand, France; clambert@chu-clermontferrand.fr (C.L.); bpereira@chu-clermontferrand.fr (B.P.); 5Université Clermont Auvergne, INSERM U1240, IMOST, F-63000 Clermont-Ferrand, France; emmanuel.moreau@uca.fr; 6Université Clermont-Auvergne, Unité INSERM 1107 Neuro-Dol, CHU Clermont-Ferrand, F-63000 Clermont-Ferrand, France; jpinguet@chu-clermontferrand.fr (J.P.); drichard@chu-clermontferrand.fr (D.R.)

The comments written by R. Otter et al., a consortium of toxicologists employed by the chemical industry, are essentially oriented towards three main themes to which we propose to provide some answers:

## 1. Description of the Care Management of the Patients Included in the Armed^®^ Study (Type and Frequency of Procedures, Composition of Medical Devices)

The medical procedures used in the NICU of Clermont-Ferrand and Lille are those usually applied in neonatal intensive care units. The medical devices (MDs) used include respiratory assistance MDs (tracheal tubes, ventilation circuits, masks, etc.), infusion and parenteral nutrition MDs (umbilical and peripherally inserted central line, catheters, infusors, extension tubings, etc.), enteral nutrition (feeding tubes, tubings) and transfusion. Many of these MDs are made of plasticized PVC. The use of these MDs and the rhythm of their replacement varies depending on a patient’s condition. 

In the context of our article, we wanted to carry out an overall assessment of patient exposure, without focusing on individual situations or kinetic monitoring. We therefore presented the exposure results by plasticizer and by type of treatment. For this purpose, we calculated an exposure dose based on various elements such as the plasticizer composition of each MD, its dimensions (contact surface with the fluid conveyed), the duration of the exposure, etc. 

The precise listing of all these factors does not seem essential to a good understanding of the study, does not bring any added value and would considerably burden down the manuscript.

## 2. Questioning the Exposure Doses Calculated with the Fick Model, in Particular the Estimated Doses for Respiratory Assistance MDs

R. Otter et al. believes that our model is unreliable, and they provide references related to other diffusion models. The proposed references are related to gas-phase diffusion models studied the migration of DINP and DINCH. However, these plasticizers are not those contained in our respiratory assistance MDs, which are plasticized with DEHP. There are several publications presenting the diffusion of DEHP from PVC to gases [1,2,3,4,5,6]. We relied on these scientific publications to apply a corrective factor to our Fick model for the MDs concerned (see page 13 of the article). Our estimation of the exposure to DEHP via respiratory assistance MDs is admittedly open to criticism, but is based on more appropriate scientific arguments than those proposed by R. Otter et al.

## 3. Risk Estimation Related to Plasticizers Contained in MDs

We will let industrial toxicologists take responsibility for what they say about the toxicity of plasticizers, especially DEHP, and the potential risk to children they could cause. We simply wish to insist on the value of biomonitoring studies for estimating the exposure risk of individuals in real situations. We nevertheless take note of the arguments of R. Otter et al. to put the plasticizers’ toxicity into perspective. 

From our point of view as clinicians, only through the multiplication of biomonitoring studies is possible to judge the consistency of results. However, it is an indisputable fact that the first 1000 days of life represent a critical period during which neonates and children are particularly vulnerable to exposure to low doses of endocrine-disrupting substances, and the consequences can be serious [7,8,9,10,11]. It is therefore important to continue to assess the exposure of patients, particularly premature babies in intensive care units who are highly exposed to potential toxic compounds leaching from medical devices. 

Field studies such as the Armed^®^ study are essential to enrich knowledge on the subject and provide all the elements aimed at securing the therapeutic care of patients. We believe that progress is still to be made to improve the quality of medical devices and to eliminate the presence of any compound that may present the slightest risk of being an endocrine disruptor.

## Data Availability

The data presented in this study are available on request from the corresponding author. The data are not publicly available due to the misuse of data dissemination that can be made regarding the plasticizer composition of the various medical devices.

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
