# Peer review of "Reply to Otter et al. Comment on “Bernard et al. Association between Urinary Metabolites and the Exposure of Intensive Care Newborns to Plasticizers of Medical Devices Used for Their Care Management. Metabolites 2021, 11, 252”"

_metabolites, 2021, doi:10.3390/metabo11090598_

Round 1

Reviewer 1 Report

I do not have concerns.

Author Response

Reviewer 1 doesn’t have any concerns.

Reviewer 2 Report

Lise Bernard et al had written a reply to the comments of Otter et al, to the manuscript "Association bteween Urinary Metabolites and the Exposure of Intensice Care Newborns to plasticizers of Medical Devices Used for Their Care Managemen".

The response to Otter's comments is adequate in form and interesting in substance.

The authors justify the approach of the manuscript and respond in three points to the comments of Otter et al. The answers are clear and, in my view, adequate

As the only criticism, perhaps Lise Bernard et al should provide some bibliographic reference to support their text.

Author Response

References were added in the manuscript at two levels in the text (see the number of added references underlined in yellow)

The manuscript was read by a native English speaker.

Reviewer 3 Report

Bernard et al. respond to the comment on their study published in Metabolites in 2021 by Otter et al. Responses are valid and merit to be published. For better readability, I would however suggest a point-by-point answer to the comments made by Otter et al.

Author Response

Thank you for your suggestion. We had initially envisioned a point-by-point response. However, this method didn’t seem to sufficiently highlight the messages we wanted to convey. In order to bring more scope to our comments, we made the collective choice to group the responses into three main themes. We would like to keep this presentation.

The manuscript was read by a native English speaker.

Reviewer 4 Report

no further comments are required. 
Authors have answered sufficiently. 

Author Response

Reviewer 2 doesn’t have any concerns.

The manuscript was read by a native english

Round 2

Reviewer 3 Report

I have no further comments.